# CCSRD: Content-Centric Speech Representation Disentanglement Learning for End-to-End Speech Translation

**Xiaohu Zhao, Haoran Sun, Yikun Lei, Shaolin Zhu, Deyi Xiong**[*]
College of Intelligence and Computing, Tianjin University, Tianjin, China
{zhaoxiaohu, hrsun, yikunlei, zhushaolin, dyxiong}@tju.edu.cn

## Abstract

Deep neural networks have demonstrated their capacity in extracting features from speech inputs. However, these features may include non-linguistic speech factors such as timbre and speaker identity, which are not directly related to translation. In this paper, we propose a content-centric speech representation disentanglement learning framework for speech translation, CCSRD, which decomposes speech representations into content representations and non-linguistic representations via representation disentanglement learning. CCSRD consists of a content encoder that encodes linguistic content information from the speech input, a non-content encoder that models non-linguistic speech features, and a disentanglement module that learns disentangled representations with a cyclic reconstructor, feature reconstructor and speaker classifier trained in a multi-task learning way. Experiments on the MuST-C benchmark dataset demonstrate that CCSRD achieves an average improvement of +0.9 BLEU in two settings across five translation directions over the baseline, outperforming state-of-the-art end-to-end speech translation models and cascaded models.

## 1 Introduction

End-to-end (E2E) speech-to-text translation (ST) aims to translate a source speech input into a target translation. Compared with traditional cascaded ST models, E2E ST models avoid the issue of error propagation and exhibit low inference latency. Recent works have made significant progress in E2E ST, which enables it to even outperform traditional cascaded systems on language pairs like En-De and En-Ru (Ye et al., 2022; Fang et al., 2022; Du et al., 2022a).

Despite the remarkable progress, E2E ST is still confronted with challenges in learning desirable

---

*Corresponding author

speech representations. Speech inputs to ST encompass not only content information that is essential for translation but also non-linguistic factors such as pitch, timbre, prosody, speaker identity and so on. As a result, the speech encoder encodes not only the content information but also a range of non-linguistic speech elements. Such non-linguistic factors may benefit translation, e.g., prosody as mentioned by (Sperber and Paulik, 2020). However, they may also introduce spurious correlations between speech inputs and target translations, undermining speech translation generalization.

To mitigate this issue, we propose to explore speech representation disentanglement learning in the context of speech translation, which aims to separate content from non-linguistic factors (e.g., prosody, timbre). Previous studies have consistently demonstrated the efficacy of representation disentanglement in improving model generalization (Chen et al., 2016; Sanchez et al., 2020; Chan and Ghosh, 2022; Mohamed et al., 2022). Given the highly complex nature of speech features and the modality gap between text and speech, we argue that representation disentanglement could enable ST to focus on content, reducing the negative influence from non-linguistic factors on speech translation modeling.

Specifically, we propose a **C**ontent-**C**entric **S**peech **R**epresentation **D**isentanglement learning framework, termed as CCSRD, for end-to-end speech translation. We re-function the original encoder as the content encoder that encodes linguistic content information contained in the speech input. We introduce an additional encoder, non-content encoder, to encode non-linguistic speech features. The decomposition of the speech input into content and non-linguistic factors for the two encoders is completed by a disentanglement module. The disentanglement module is trained in a multi-task learning way, which leverages three

tasks: a cyclic reconstruction task to reduce the mutual information between content representation and non-content representation, a feature reconstruction task to ensure the retention of speech information, and a speaker classifier task to guide the training of the non-content encoder. Additionally, we also employ a masking strategy to further improve disentanglement.

It is noteworthy that our method does not require transcription to achieve representation disentanglement. Therefore it can be used in scenarios that are short of transcription data or do not have transcription data at all. Our work is hence significantly different from most previous works that heavily depend on transcription for speech translation.

Although disentangled speech representation learning is not new in the community of speech processing (Xie et al., 2021; Wang et al., 2021; Abeßer and Müller, 2021), to the best of our knowledge, this is the first attempt to learn disentangled speech representations for end-to-end speech translation. In a nutshell, our contributions are listed as follows.

- We propose CCSRD for end-to-end ST, which learns speech representation disentanglement to separate content from non-linguistic features.

- The proposed disentanglement module consists of cyclic reconstruction, feature reconstruction and speaker classifier. It does not require any transcription data for disentanglement learning.

- We conduct extensive experiments on the MuST-C benchmark with five language pairs. CCSRD achieves an average improvement of +0.9 BLEU in a setting without using any transcription data and +0.9 BLEU in ST with the multi-task (MTL) setting using transcription data.

## 2 Related Work

**End-to-End Speech Translation** To avoid error propagation in cascaded ST and reduce inference latency, Bérard et al. (2016) and Weiss et al. (2017) propose end-to-end ST that directly translates speech in the source language into text in the target language, without relying on the intermediate transcriptions. However, due to the inherent complexity and variation of speech signals and the scarcity of high-quality E2E ST data, achieving satisfactory performance remains challenging. Over the years, a variety of approaches have been proposed to address these issues, such as pre-training (Wang et al., 2020b; Tang et al., 2021b; Dong et al., 2021), multi-task learning (Vydana et al., 2021; Ye et al., 2021; Tang et al., 2022), data augmentation (Jia et al., 2019; Lam et al., 2022), contrastive learning (Li et al., 2021; Ye et al., 2022) and knowledge distillation (Tang et al., 2021a; Zhao et al., 2021). Most of these approaches focus on using the transcription data in speech data triplets to perform MT/ASR tasks, pretrain model components and mitigate the modality gap between speech and text. Significantly different from them, we attempt to improve translation quality by efficiently exploring the speech representation disentanglement to learn content-centric speech representations for ST.

**Representation Disentanglement** Representation disentanglement refers to a learning paradigm in which models represent input signals through multiple separated dimensions or embeddings. Therefore, it is always advantageous in obtaining representations that carry certain attributes or extract discriminative features. Reconstruction based training (Gonzalez-Garcia et al., 2018; Zhang et al., 2019; Bertoin and Rachelson, 2022) is widely adopted in disentanglement learning and used to obtain disentangled representations. The application of representation disentanglement is extensive, including speech (Chan and Ghosh, 2022; Chan et al., 2022), computer vision (Gonzalez-Garcia et al., 2018; Lee et al., 2021) and natural language precessing (Bao et al., 2019; Cheng et al., 2020). Since speech often contains multiple factors, disentangled representation learning provides a way to extract different representations for different tasks like voice conversion (Du et al., 2022b), automatic speech recognition (Chan and Ghosh, 2022) and speaker recognition (Kwon et al., 2020). Our approaches are partially motivated by these efforts but are significantly different from them in two aspects. First, we make the first step to use representation disentanglement during the training stage of E2E ST and propose CCSRD. Second, we focus on the quality of the content representation for ST, removing the additional encoder and disentanglement module during inference, while the previous works focus on the disentanglement of different speech factors for various speech tasks.

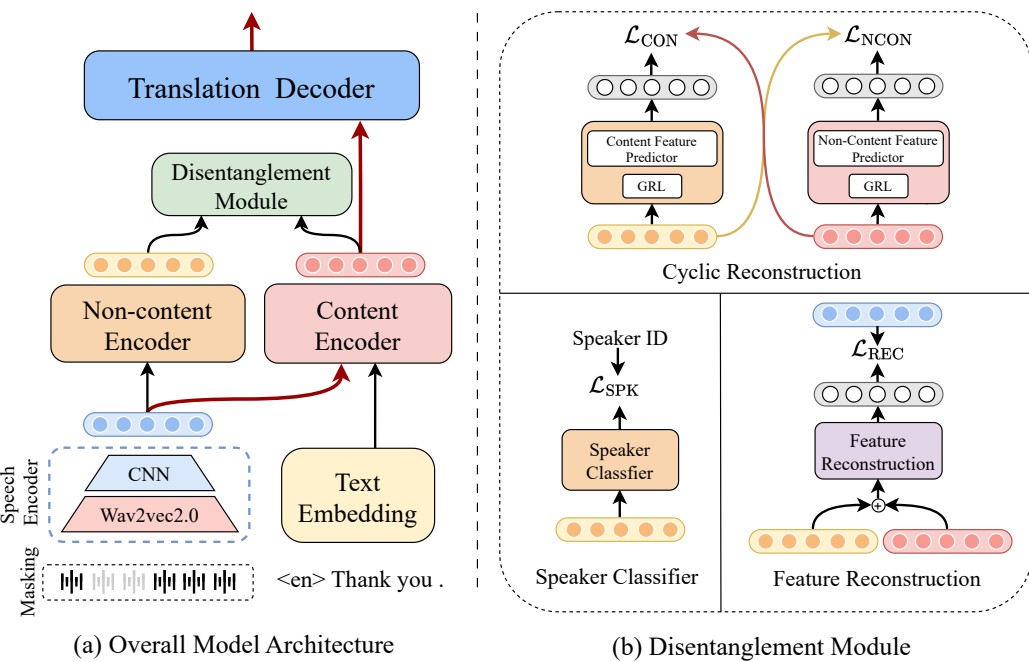

(a) Overall Model Architecture       (b) Disentanglement Module

Figure 1: Diagram of the proposed content-centric speech representation disentanglement learning framework for E2E ST. **Left**: The overall architecture, where the text embedding layer is only used during the training under the MTL setting. The red arrow lines denote the data flow at inference. **Right**: Details of disentanglement module. It has three parallel tasks: cyclic reconstruction, speaker classifier and feature reconstruction.

## 3 CCSRD

In this section, we first introduce the overall model architecture of CCSRD and subsequently elaborate disentanglement learning, training and inference of CCSRD.

### 3.1 Model Architecture

Our E2E ST model adopts the encoder-decoder ST backbone. As shown in Figure 1, it consists of five essential components: speech encoder, content encoder, non-content encoder, disentanglement module and translation decoder.

**Speech Encoder** is to extract low-level features from speech signals. It contains Wav2vec2.0 (Baevski et al., 2020) and two additional convolutional layers, which are added to shrink the extracted speech features by a factor of 4.

**Content Encoder** employs the same configuration as the original Transformer (Vaswani et al., 2017) encoder. The input of the content encoder is the output of the speech encoder for both ASR and ST tasks while the embeddings of the transcription are for MT task. The content encoder is trained to learn decomposed content representations.

**Non-content Encoder** employs the same configurations as the content encoder. The input of the non-content encoder is the output of the speech

encoder. The non-content encoder is trained to learn non-linguistic representations separated from content representations.

**Translation Decoder** employs the base configurations as the original Transformer (Vaswani et al., 2017) decoder, which is shared by ASR, MT and ST. It yields either speech transcription or target translation based on the output of the content encoder.

**Disentanglement Module** is composed of four networks: content feature predictor network that predicts content representations from non-content features, and non-content feature predictor network that predicts non-content representations from content features, speaker classifier network that predicts the speaker IDs based on non-content representations and feature reconstruction network that predicts speech features from the speech encoder based on both content and non-content representations.

The speech encoder, content encoder and translation decoder are the same as those used by Ye et al. (2021). The non-content encoder and disentanglement module are additionally incorporated into CCSRD, which are removed during the inference stage.

The training data of ST usually contains speech-

transcription-translation triples, which can be denoted as $D = \{(\mathbf{s}, \mathbf{x}, \mathbf{y})\}$. With these training instances, E2E ST can be trained in two ways. Similar to standard neural machine translation, it can be trained in the way of being treated as a single ST task:

$$\mathcal{L}_{\text{ST}} = -\sum_{(\mathbf{s},\mathbf{y})\in D} \log P(\mathbf{y}|\mathbf{s}) \quad (1)$$

As $|D|$ is usually not large, E2E ST is often trained in a multi-task learning way (Ye et al., 2021). Additional ASR and MT tasks are incorporated into the training of ST with speech-transcription pairs $\{(\mathbf{s}, \mathbf{x})\}$ and transcription-translation pairs $\{(\mathbf{x}, \mathbf{y})\}$ to facilitate knowledge transfer:

$$\mathcal{L}_{\text{ASR}} = -\sum_{(\mathbf{s},\mathbf{x})\in D} \log P(\mathbf{x}|\mathbf{s}) \quad (2)$$

$$\mathcal{L}_{\text{MT}} = -\sum_{(\mathbf{x},\mathbf{y})\in D} \log P(\mathbf{y}|\mathbf{x}) \quad (3)$$

### 3.2 Speech Representation Disentanglement Learning

In the training stage of CCSRD, we attempt to decompose representations from the speech encoder into two distinct components: content representations that are essential for translation, and non-content representations that encode non-linguistic speech factors. To achieve this, we propose three training strategies: (1) encouraging the content encoder to encode only linguistic content information, (2) encouraging the non-content encoder to encode non-linguistic speech factors other than linguistic content, (3) minimizing the mutual information between the content and the non-content representations. For the first training strategy, we mainly force the content encoder to learn content-centric representations to reduce ST loss. For the second and third strategies, we propose cyclic reconstruction, feature reconstruction, and speaker classifier tasks.

**Cyclic Reconstruction** To effectively disentangle speech representations, we employ the cyclic reconstruction method proposed by Bertoin and Rachelson (2022) to reduce the mutual information between the content and non-content representations in an unsupervised learning setting. Specifically, after the extraction of content and non-content representations from the content and non-content encoder respectively, we stack two sub-networks: the content feature predictor $\phi_{\text{content}}$ and non-content

feature predictor $\phi_{\text{non-content}}$ to cyclically reconstruct the content and non-content representations. Particularly, we train the content and non-content feature predictor with the following two reconstruction losses $\mathcal{L}_{\text{CON}}$ and $\mathcal{L}_{\text{NCON}}$ respectively:

$$\mathcal{L}_{\text{CON}} = \sum_{i=1}^{|D|} ||\mathbf{H}_i^c - \phi_{\text{content}}(\mathbf{H}_i^s)||_2^2 \quad (4)$$

$$\mathcal{L}_{\text{NCON}} = \sum_{i=1}^{|D|} ||\mathbf{H}_i^s - \phi_{\text{non-content}}(\mathbf{H}_i^c)||_2^2 \quad (5)$$

where $\mathbf{H}_i^c$ and $\mathbf{H}_i^s$ represent the representation for $i^{th}$ speech input from content and non-content encoder respectively. These two sub-networks are connected to the overall architecture through Gradient Reversal layers (GRLs) (Ganin et al., 2016). GRLs invert the gradient sign during the backward pass, thus pushing the parameters to maximize the losses. As such, the model is constrained to produce content and non-content representations that are hard to reconstruct each other, which means minimal information is shared between the two types of representations (as their distance is maximized by Eq. (4) and (5)).

**Feature Reconstruction** We employ the reconstruction feature predictor network to predict the original speech features output by the speech encoder based on non-content representations learned by the non-content encoder and content representations learned by the content encoder. The training objective for this sub-network is defined as:

$$\mathcal{L}_{\text{REC}} = \sum_{i=1}^{|D|} ||\mathbf{H}_i - \phi_{\text{rec}}(\mathbf{H}_i^c \oplus \mathbf{H}_i^s)||_2^2 \quad (6)$$

where $\mathbf{H}_i$ represents the representation for $i^{th}$ speech input output from the speech encoder, $\phi_{\text{rec}}$ is the feature reconstruction network which tries to recover the speech representation from both content representation $\mathbf{H}_i^c$ and non-content representation $\mathbf{H}_i^s$.

**Speaker Classifier** Minimizing the mutual information between the content and non-content representations may be not sufficient as the non-content encoder is not fully constrained and goal-oriented. To address this issue, we make full use of speaker ID information in the dataset. The speaker ID can be considered as a label for non-linguistic factors. We believe that the non-content encoder should fully model this information. Modeling non-linguistic factors would benefit the decoupling of

**Algorithm 1:** CCSRD in the MTL setting

**Input** : A batch of training set (**s**, **x**, **y**)
**Output**: Loss

1 **while** *not converged* **do**
2     get $\mathbf{s}'$ from **s** by applying the masking strategy;
3     get **H** for $\mathbf{s}'$ from the speech encoder;
4     get $\mathbf{H}^s$ and $\mathbf{H}^c$ for $\mathbf{s}'$ from the non-content and content encoder;
5     get $\mathcal{L}_{\text{SRD}}$ in Eq. (9) from the disentanglement module with the input **H**, $\mathbf{H}^s$ and $\mathbf{H}^c$;
6     do ST task with $\mathbf{H}^c$ and **y**, get $\mathcal{L}_{\text{ST}}$;
7     do ASR task with $\mathbf{H}^c$ and **x**, get $\mathcal{L}_{\text{ASR}}$;
8     get $\mathbf{H}^c$ for **x** from the content encoder;
9     do MT task with $\mathbf{H}^c$ and **y**, get $\mathcal{L}_{\text{MT}}$;
10     get $\mathcal{L}_{\text{MTL}}$ in Eq. (10);
11 **end**

non-linguistic information from content information. We train the speaker classifier network with the following objective:

$$\mathcal{L}_{\text{SPK}} = -\sum_{i=1}^{|\boldsymbol{D}|} \log P\left(\mathbf{spk}_i \mid \mathbf{H}_i^s\right) \quad (7)$$

where $\mathbf{spk}_i$ represents the speaker ID for the $i^{th}$ speech input.

### 3.3 Masking Strategy

Recent studies have demonstrated that the masking operation can significantly improve the robustness of models and benefit speech representation disentanglement in multiple speech tasks (Chan and Ghosh, 2022; Lin et al., 2023). As shown in Figure 1, during the training stage, we modify the input waveform **s** of the speech encoder to mask consecutive segments and obtain the masked waveform $\mathbf{s}'$. Then we take $\mathbf{s}'$ as the input to the model. Specifically, for each speech input, it is chosen to be masked with a probability $p$. The selected speech input is then masked for at least $n$ spans, and each span contains at least $m$ consecutive frames. In our experiments, we set $p$ to 0.75, $n$ to 2 and $m$ to 3600.

### 3.4 Training and Inference

CCSRD can be trained either in the ST task setting or in the multi-task setting with both MT and ASR tasks. In the ST task setting, the overall training objective is:

$$\mathcal{L} = \mathcal{L}_{\text{ST}} + \mathcal{L}_{\text{SRD}} \quad (8)$$

where

$$\mathcal{L}_{\text{SRD}} = \mathcal{L}_{\text{CON}} + \mathcal{L}_{\text{NCON}} + \mathcal{L}_{\text{REC}} + \mathcal{L}_{\text{SPK}} \quad (9)$$

In the multi-task setting, the training process overall training objective is:

$$\mathcal{L}_{\text{MTL}} = \mathcal{L}_{\text{ST}} + \mathcal{L}_{\text{ASR}} + \mathcal{L}_{\text{MT}} + \mathcal{L}_{\text{SRD}} \quad (10)$$

The training process for the multi-task setting is demonstrated in Algorithm 1. During the inference stage, only the speech encoder, content encoder and translation decoder are used, which is consistent with previous works. Hence, our method does not introduce any additional inference latency compared with previous methods.

## 4 Experiments

### 4.1 Datasets

We conducted experiments on the widely-used MuST-C multilingual speech translation dataset (Di Gangi et al., 2019). We carried out experiments on English-to-German (DE), English-to-Spanish (ES), English-to-Russian (RU), English-to-French (Fr) and English-to-Italian (It). *dev* was used to develop and analyze our approaches, *tst-COMMON* was used for testing. See Appendix A for detailed dataset statistics.

### 4.2 Settings

**Model Configurations** We used Wav2vec2.0 in the speech encoder, which is only pretrained on audio data from Librispeech (Panayotov et al., 2015) without performing any downstream finetuning. The kernel, stride and hidden size of the two CNN layers stacked over Wav2vec2.0 were set to 5, 2 and 512. The used content encoder and decoder follow the base configuration of Transformer, with 6 layers, 512 hidden sizes and 8 attention heads. The non-content encoder follows the same configuration as the content encoder. For the network architecture of the disentanglement module in the experiments, both the content feature predictor network and non-content feature predictor network consist of three fully connected (FC) layers followed by ReLU activation and one FC layer followed by Tanh activation. The feature reconstruction network utilizes the same architecture as the feature predictor network, except for the different

| Models | En-De | En-Es | En-Ru | En-Fr | En-It | Avg. |
|---|---|---|---|---|---|---|
| *Training in the ST setting w/o using transcription data* | | | | | | |
| Fairseq ST (Wang et al., 2020a) | 22.7 | 27.2 | 15.3 | 32.9 | 22.7 | 24.2 |
| Self-training (Pino et al., 2020) | 25.2 | - | - | 34.5 | - | - |
| SpecRec (Chen et al., 2021) | 20.8 | 25.3 | 13.1 | 30.3 | 20.5 | 22.0 |
| Revisit ST (Zhang et al., 2022) | 23.0 | 28.0 | 15.6 | 33.5 | 23.5 | 24.7 |
| W2V2-Transformer (Fang et al., 2022) | 24.1 | 29.4 | 16.3 | 35.0 | 24.8 | 25.9 |
| ST baseline | 24.5 | 29.4 | 16.1 | 34.8 | 24.9 | 25.9 |
| CCSRD | **25.4**** | **30.2**** | **16.9**** | **35.8**** | **25.8**** | **26.8** |
| *Training in the MTL setting w/ using transcription data* | | | | | | |
| XSTNet (Ye et al., 2021) | 25.5 | 29.6 | 16.9 | 36.0 | 25.5 | 26.7 |
| SATE (Xu et al., 2021) | 25.2 | - | - | - | - | - |
| Mutual-learning (Zhao et al., 2021) | - | 28.7 | - | 36.3 | - | - |
| STEMM (Fang et al., 2022) | 25.6 | 30.3 | 17.1 | 36.1 | 25.6 | 26.9 |
| ConST (Ye et al., 2022) | 25.7 | 30.4 | 17.3 | 36.8 | 26.3 | 27.3 |
| MTL baseline | 25.5 | 30.0 | 17.0 | 36.0 | 25.7 | 26.8 |
| CCSRD | **26.1**** | **31.0**** | **17.8**** | **37.1**** | **26.4**** | **27.7** |

Table 1: BLEU scores of different models on the MuST-C tst-COMMON set. "ST baseline" and "MTL baseline" are the implemented strong baselines using the same architecture as our model, excluding the disentanglement module and non-content encoder. ** denotes that the improvements over W2V2-Transformer baseline is statistically significant ($p < 0.01$).

input dimensions, since the input of the network is the concatenation of content and non-content representation. The classifier network comprises three FC layers with ReLU activation, an adaptive average pooling layer and a log softmax layer.

**Experiment Details** We used the raw 16-bit 16 kHz mono-channel audio wave as speech input and removed utterances of which the duration is longer than 300K frames. For each translation direction, we used a unigram SentencePiece (Kudo and Richardson, 2018) model to learn a vocabulary on the text data from the dataset, which is the same as Ye et al. (2021, 2022)'s setup. For evaluation, we computed case-sensitive detokenized BLEU using sacreBLEU (Post, 2018) on MuST-C tst-COMMON set. Appendix B contains more detailed settings.

**Baselines** We compared our model with multiple strong E2E ST baselines including: (1) Fairseq ST (Wang et al., 2020a), (2) Self-training (Pino et al., 2020), (3) SpecRec (Chen et al., 2021), (4) Revisit ST (Zhang et al., 2022), (5) W2V2-Transformer (Fang et al., 2022), (6) XSTNet (Ye et al., 2021), (7) SATE (Xu et al., 2021), (8) Mutual-learning (Zhao et al., 2021), (9) STEMM (Fang et al., 2022) and (10) ConST (Ye et al., 2022). Additionally,

we compared with a strong baseline "ST baseline" that uses the same neural architecture with W2V2-Transformer (excluding the proposed non-content encoder and disentanglement module). We also compared against a strong "MTL baseline" that uses the same neural architecture as the ST baseline but is trained in the MTL setting, which is the same as XSTNet (Ye et al., 2021).

### 4.3 Main Results

**Comparison to End-to-End Baselines** We compared our model with baselines for five language pairs on the MuST-C benchmark dataset. Results are shown in Table 1. In the ST setting that does not use transcriptions in the MUST-C data triples, our model achieves a substantial improvement of 0.9 BLEU over the ST baseline on average and outperforms the strongest baseline, W2V2-Transformer, in all translation directions. Since MTL models have achieved state-of-the-art results in recent studies, we also implemented the proposed CCSRD in the MTL setting that employs transcriptions to perform MT and ASR tasks. Our model achieves a 0.9 BLEU improvement over the MTL baseline, and even the strong model, ConST. These validate the effectiveness of our proposed approach in enhanc-

| Models | En-De | En-Ru |
|--------|-------|-------|
| **Cascaded** | | |
| Espnet | 23.6 | 16.4 |
| (Ye et al., 2021) | 25.2 | 17.0 |
| **End-to-end** | | |
| CCSRD | **26.1** | **17.8** |

Table 2: Comparison to the cascaded baselines on the MuST-C En-De and En-Ru tst-COMMON set. "Cascaded" is the implemented strong cascaded baselines.

| Models | En-De | En-It |
|--------|-------|-------|
| CCSRD | **25.4** | **25.8** |
| w/o masking | 25.1 | 25.5 |
| w/o $\mathcal{L}_{\mathrm{CON}}$ and $\mathcal{L}_{\mathrm{NCON}}$ | 24.7 | 25.1 |
| w/o $\mathcal{L}_{\mathrm{SPK}}$ and $\mathcal{L}_{\mathrm{REC}}$ | 24.5 | 24.9 |

Table 3: Ablation study results on the MuST-C En-De tst-COMMON set.

ing ST performance using speech representation disentanglement. Notably, even not being trained in the MTL setting, our model achieves the same BLEU score as the baseline model trained in the MTL setting, but with a reduced training time.

**Comparison to Cascaded Baselines** We compared our method with several cascaded baselines. Espnet (Inaguma et al., 2020) and the model proposed by Ye et al. (2021) are two strong cascaded systems trained using MuST-C and external ASR and MT data. From Table 2, we find that as an end-to-end model, our model outperforms these strong cascaded models even not using any external data.

### 4.4 Ablation Study

To gain a deep understanding of the effect of components deployed in our proposed model, we conducted an ablation study by progressively removing the masking strategy, the cyclic reconstruction loss $\mathcal{L}_{\mathrm{CON}}$ and $\mathcal{L}_{\mathrm{NCON}}$, and the speaker classifier loss $\mathcal{L}_{\mathrm{SPK}}$ and feature recontruction loss $\mathcal{L}_{\mathrm{REC}}$. Results are shown in Table 3, which indicate that the masking strategy contributes to an average improvement of 0.3 BLEU. Besides, CCSRD without masking strategy can significantly enhance the translation performance, with an improvement of 0.6 BLUE achieved both in the En-De and En-It translation directions. Moreover, the cyclic reconstruction task plays an extremely important role in speech representation disentanglement, without which the per-

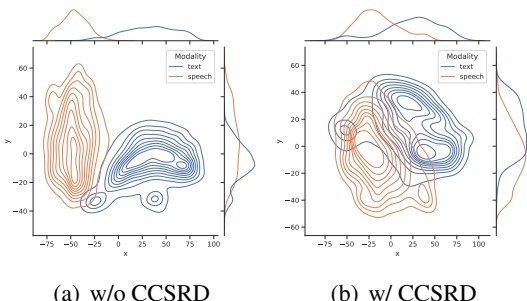

| (a) w/o CCSRD | (b) w/ CCSRD |
|---|---|

Figure 2: Bivariate KDE contour of the speech and text representations. Blue curves are text representations while orange curves represent speech representations. Samples are drawn from the MuST-C En-De tst-COMMON set.

| Models | w/o Noise | w/ Noise |
|--------|-----------|----------|
| ST baseline | 24.5 | 24.2 |
| CCSRD | **25.4** | **25.3** |

Table 4: BLEU scores on the MuST-C En-De tst-COMMON set while artificially noisy is introduced during training.

formance drops substantially. Experimental results convincingly demonstrate the effectiveness of these approaches in improving ST performance.

## 5 Analysis

### 5.1 Can CCSRD Learn Content-Centric Representations?

To empirically demonstrate the effectiveness of our model in learning content-centric representations, we plot the bivariate kernel density estimation (KDE) (Parzen, 1962) contour of speech and text dim-reduced representations to visualize their distributions Figure 2, where t-SNE (Van der Maaten and Hinton, 2008) is used to reduce the dimension of representations into 2D. Ideally, if the representations of speech have less non-linguistic information, their KDEs will be similar to the representations of text, resulting in the contour lines overlapping as much as possible. As illustrated in Figure 2, without CCSRD, the overlap between speech representation distribution and text representation distribution is small. In contrast, when CCSRD is applied, representations of different modalities become closer compared with those learned by the baseline. This suggests that speech representations contain more content information similar to that of text representations, which is beneficial to ST.

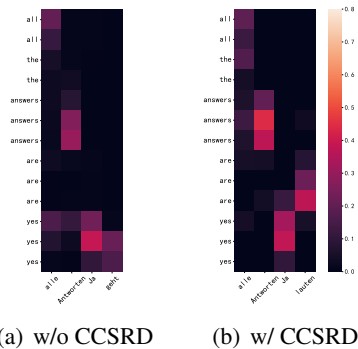

| (a) w/o CCSRD | (b) w/ CCSRD |

Figure 3: Visualization of cross-attention learned by CCSRD vs. ST baseline by averaging all heads of the last cross-attention layer.

We also conducted experiments to analyze the disentanglement ability of CCSRD. We trained CCSRD and ST baseline on data with artificially-generated noise and compared their performance. Noise data were obtained by adding part of the information of another speech to the original speech input. Specifically, for a speech input $s_i$, we randomly selected another speech $s_j$ from the training set as noise, and add it to the $s_i$ according to the preset weight of 0.15 to obtain a new noisy data: $s_i' = s_i + 0.15 * s_j$. Results of these experiments are shown in Table 4. We observe that the performance of CCSRD degrades slightly while the performance of the ST baseline drops substantially. It validates that our method is able to achieve successful disentanglement and that the content encoder is capable of learning high-quality content-centric representations.

To further compare the efficacy of our approach against the baseline, we visualize the cross-attention matrices of the ST baseline and CCSRD in Figure 3. We obtain the attention matrix by averaging all heads of the last cross-attention layer. Our model confidently aligns target tokens to their corresponding speech frames respectively as in Figure 3, where the baseline model yields an incorrect translation with inappropriate attention weights.

## 5.2 Dose External MT Data Further Improve CCSRD ?

Many previous studies regard "leveraging external MT data" to be one of the advantages of their models and achieve better performance in the MTL setting. Therefore we further investigated CCSRD trained with external WMT16 En-De dataset. Results are shown in Table 5. We observe that CCSRD

| External MT | MTL baseline | CCSRD |
|:-:|:-:|:-:|
| ✗ | 25.5 | **26.0** |
| ✓ | 27.2 | **28.1** |

Table 5: BLEU scores of CCSRD vs. MTL baseline on the MuST-C En-De tst-COMMON set while external MT data is used during training.

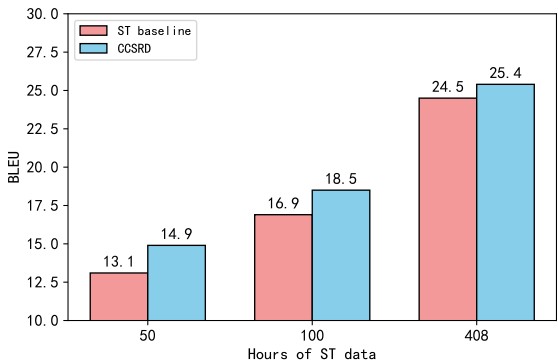

Figure 4: BLEU scores on the MuST-C En-De tst-COMMON set when the amount of speech data used for training is varying from 50K to 408K hours.

trained with external MT data in the MTL setting achieves a further improvement of 2.1 BLUE while the MTL baseline obtains an improvement of 1.7 BLUE. We conjecture that with the external MT data for training the MT task in the MTL setting, CCSRD is able to learn more on content-to-target-translation alignment and hence to generate better translations.

## 5.3 CCSRD in Low-Resource Scenarios

We extend the proposed CCSRD to ST in low-resource scenarios in the ST setting to investigate if it is still able to improve translation quality. We conducted experiments using different hours of speech data from the training dataset, simulating low-resource conditions. The results are shown in Figure 4. We reduced the ST data to 50 and 100 hours, corresponding to around 25K and 50K sentences. We observe that CCSRD is particularly helpful when the amount of speech data is small.

## 6 Conclusions

In this work, we have presented CCSRD, an E2E ST framework that explores speech representation disentanglement learning to capture content representations for ST. Experimental results validate the effectiveness of the proposed framework under both the ST task setting and the multi-task setting.

In-depth analyses demonstrate that CCSRD is capable of disentangling linguistic content from non-linguistic speech factors. We would like to employ more speech factors to guide the training of the non-content encoder and explore the disentangled non-content representations for improving ST in the future.

## Limitations

Although the proposed method facilitates ST to learn content-centric representation, and obtains significant improvements over previous methods, it still has some limitations: (1) We need labels of non-linguistic speech factors (e.g., speaker IDs, prosody labels) to guide the training of the non-content encoder. (2) It is difficult to analyze the non-content representation since they may be related to many different speech factors. (3) Non-content representations could also be used to improve ST, which we leave to our future work.

## Ethics Statement

This work presents a framework and training strategies to help the model learn content-centric representations for speech translation. We evaluated our method on the widely-used standard benchmark dataset and did not introduce additional data that may cause ethics issues.

## Acknowledgments

The present research was supported by the Natural Science Foundation of Xinjiang Uygur Autonomous Region (No. 2022D01D43) and the Key Research and Development Program of Yunnan Province (No. 202203AA080004). We would like to thank the anonymous reviewers for their insightful comments.

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

## A  Statistics of all datasets

| Languages | Hours | Sents |
|-----------|-------|-------|
| **En→De** | 408 | 234K |
| **En→Es** | 504 | 270K |
| **En→Fr** | 492 | 280K |
| **En→It** | 465 | 258K |
| **En→Ru** | 489 | 270K |

Table 6:  Statistics of the used MuST-C datasets.

## B  Experimental Details

**Training and Implementation Details** We implemented our model based on the fairseq toolkit and trained all models on 8 A6000 GPUs. During the training stage, we used Adam optimizer with $\beta_1 = 0.9$, $\beta_2 = 0.98$, and learning rate = $1e^{-4}$ with warmup 25k steps during training. During the inference stage, we saved the checkpoints with the highest BLEU and averaged the last 10 checkpoints. For decoding, following previous studies (Ye et al., 2022), we used a beam size of 10 and a length penalty of 0.7 for German, 0.1 for Spanish, 0.5 for Italian and 0.4 for Russian. For the experiments in Table 5, we followed the previous works (Ye et al., 2021, 2022) and used the WMT16 En-De datasets as the external MT dataset, which contains 4.6M sentences.

**Baseline Model Details** In Table 1, we compared our method with end-to-end baseline models under the setting of no transcription data being used (i.e., only training the ST task):

- Fairseq ST (Wang et al., 2020a): a reimplemented model based on the Fairseq tooklit, which is trained with only the ST task data.

- Self-training (Pino et al., 2020): a model trained with pseudo-labels.

- SpecRec (Chen et al., 2021): a model trained with a spectrogram reconstruction technique.

- Revisit ST (Zhang et al., 2022): a model that trained with several techniques like parameterized distance penalty and CTC-based regularization.

- W2V2-Transformer (Fang et al., 2022): a model that has the same structure as our ST baseline.

We also compared our method against the following baseline models under the MTL setting using transcription data, (i.e., using the transcription data).:

- XSTNet (Ye et al., 2021): a model that has the same structure as W2V2-Transformer but is adopted to the multi-task fine-tuning strategy.

- STEMM (Fang et al., 2022): a model that bridges the modality representation gap by mixing up the speech representation sequences and text transcription embedding sequences.

- ConST (Ye et al., 2022): a model that applies contrastive learning to bridge the modality gap between speech and transcriptions.

- Mutual-learning (Zhao et al., 2021): a model that introduces a mutual-learning paradigm to iteratively learn and share the knowledge between MT and ST task.

- SATE (Xu et al., 2021): a model that leverages an adapter to incorporate pre-trained ASR and MT models into E2E ST.

## C  Case Analysis

We present two translation examples yielded by CCSRD in comparison to those by the ST baseline model in Table 7. We observe that ST baseline cannot accurately translate some phrases, whereas CCSRD successfully conveys the meaning, and translates more accurately than ST baseline. This improvement might be due to the ability of CCSRD in capturing content representations.

| Models | |
|---|---|
| **Example 1** | |
| Ref. | **src:** And the motives of online criminals are very easy to understand. 
 **tgt:** Und die Motive von Online-Kriminellen sind sehr leicht zu verstehen. |
| ST baseline | **tgt:** Und die Motive von Online-erbrechern ist sehr leicht zu verstehen. |
| CCSRD | **tgt:** Und die Motive von Online-Kriminellen sind sehr leicht zu verstehen. |
| **Example 2** | |
| Ref. | **src:** If we want this institution to work for us, we're going to have to make bureaucracy sexy. 
 **tgt:** Wenn wir wollen, dass diese Institution für uns arbeitet, müssen wir Bürokratie sexy machen. |
| ST baseline | **tgt:** Wenn wir wollen, dass diese Einrichtung für uns funktioniert, müssen wir die Bürokratie sexy machen. |
| CCSRD | **tgt:** Wenn wir wollen, dass diese Institution für uns arbeitet, müssen wir Bürokratie sexy machen. |

Table 7: Translation examples from the En-De tst-COMMON set, yielded by the ST baseline and CCSRD. Underlined fragments are grammatically incorrect or inaccurate translations.