# OpenReview forum: "CCSRD: Content-Centric Speech Representation Disentanglement Learning for End-to-End Speech Translation"
_EMNLP/2023/Conference — EMNLP 2023 Findings_

### Official Review · Reviewer_fmsg · 2023-08-04

**Typos Grammar Style And Presentation Improvements:** Table 4
**Soundness:** 3

**Excitement:**

4: Strong: This paper deepens the understanding of some phenomenon or lowers the barriers to an existing research direction.

**Paper Topic And Main Contributions:**

This paper proposes an improved speech-text translation system. By incorporating speech representation disentanglement during the training phase, an improvement of 0.9 BLEU is observed on the evaluation set. As speech translation is a typical NLP application, the topic is well fit to EMNLP.

**Questions For The Authors:**

A. Speech representation disentanglement involves additional multi-task learning. MTL could increase the training complexity, making unstable computation of the loss function. What are the key techniques to make the training stable?

B. Speech representation disentanglement is only performed in training stage. As the training procedure becomes complicated with MTL, what is the main benefits of this method over simply adding more training data (both speech and text)?

C. Section 5.1 and Table 4: My understanding on the “noise” is other speech. What exactly the noise source? Is the source from a single speech, babble noise, or other noise? One type of noise is not representative.  I suggest to add different types of noise in the experiments such as street noise or car engine noise.

D. Comment: My view is that the speech representation disentanglement is more useful in speech-to-speech translation. The non-content representation obtained from disentanglement process can be applied to speech generation modules to improve the generated speech.

**Reasons To Accept:**

This paper designs a speech representation disentanglement process for speech translation. In a setting of speech-to-text translation, the improved BLEU score given the same amount of training data supports the soundness of the design.

**Reasons To Reject:**

Speech representation disentanglement in speech-to-text translation is a double-edged sword. On one hand, while applying speech representation disentanglement does improve content representation and hence translation performance in terms of BLEU score, additional multi-task learning and modules complicate the training process. Given the model structure unchanged during training and inference, applying more training data is more straightforward for performance improvement, as shown in Table 5 and Figure 4.

On the other hand, yet another valuable part of speech representation disentanglement is the learnt non-content representation. The non-content representation represents the speaking style thus cannot be learnt from text-only feature. Previous works show promising results on voice cloning and generation with the disentangled non-content representation, in which these tasks are more relevant to speech-to-speech translation.  However, this work is focus on speech-to-text translation and does not use the non-content representation during inference, leaving the non-content representation untested. The performance of the non-content representation is unknown, making an incompleteness of this work.

**Reproducibility:**

3: Could reproduce the results with some difficulty. The settings of parameters are underspecified or subjectively determined; the training/evaluation data are not widely available.

**Reviewer Confidence:**

3: Pretty sure, but there's a chance I missed something. Although I have a good feel for this area in general, I did not carefully check the paper's details, e.g., the math, experimental design, or novelty.

---

> ### Author Rebuttal · Authors · 2023-08-29
>
> **We sincerely thank you for your detailed review and feedback. We will strive to integrate your constructive feedback into our paper.**
>
> **Reject_Reason1:** It is straightforward to use more training data for performance improvement.
>
> **Answer for the Reject_Reason1:** It is important to note that E2E ST data are notoriously limited and expensive to collect. The benchmark dataset MuST-C used in the paper is currently the largest speech translation dataset available to the best of our knowledge. It would be impractical to enhance translation performance just with “more training data”. Nevertheless, our method demonstrates the ability to achieve satisfactory performance even in these challenging scenarios. Moreover, it is worth noting that our method consistently achieves higher performance when supplemented with "more training data."
>
> **Reject_Reason2:** The performance of the non-content representation is unknown.
>
> **Answer for the Reject_Reason2:** We appreciate this comment. However, analyzing non-content representations poses challenges due to some objective reasons. Previous studies of speech representation disentanglement have primarily focused on specific factors within non-content information, such as emotion or rhythm, which are easier to analyze. In contrast, our method aims to disentangle the entire non-content information, which comprises numerous factors. Consequently, conducting detailed analysis on such representations becomes inherently difficult. To achieve a more profound analysis of non-content information, it may be necessary to disentangle different factors effectively. However, this would require the inclusion of additional modules, which could lead to significant increase in computational and inference time.
>
> **Question_A:** The key techniques to make the training stable.
>
> **Answer for the Question_A:** First, the backbone utilized in our paper, XSTNet, is a widely adopted framework and demonstrates stability when incorporating additional losses in previous studies. Second, we freeze the parameters of the speech encoder during the backward of $\mathcal{L}_{SRD}$, which prevents confusion in the training direction of the speech encoder that may arise from conflicting guidance from the two encoders. Third, the disentanglement method we employ is inherently stable and has been proposed under a multi-task framework. Thanks for this question. We’ll provide a detailed discussion on this in the new version.
>
> **Question_B:** The main benefits of this method over simply adding more training data.
>
> **Answer for the Question_B:** First, as mentioned in the Answer for the Reject_Reason1, speech translation data are notoriously limited and even cannot be obtained. Second, it is important to highlight that our method demonstrates improved performance while additional training data are incorporated. Third, our paper presents a robust approach to tackling the issue of non-content information, which has been identified as a bottleneck that hampers the performance of translation systems in real-world applications.
>
> **Question_C:** The noise source and the different types of noise.
>
> **Answer for the Question_C:** The noise source is the other speech in training data, simulating noisy background environment. For instance, let S1 and S2 denote the embedding of two audio inputs, we obtain the noisy input for $S_1$ by adding the embedding of $S_2$ with a weight 0.1, resulting in $S_1’ = S_1 + 0.1S_2$. Thank you for your suggestion on adding different types of noise. We will perform such an experiment in the next version.
>
> **Question_D:** The non-content representation obtained from disentanglement process can be applied to speech generation modules to improve the generated speech.
>
> **Answer for the Question_D:** We appreciate this comment and acknowledge that disentanglement learning is useful in speech-to-speech translation. However, it is important to note that speech disentanglement is also vital in speech-to-text translation. The robustness of current ST models in unseen speakers and varied environments is still very poor. In our paper, we propose a robust approach to tackling the issue that non-content information unduly influences translation, which has been shown to be a bottleneck restricting translation systems in real-world applications.
>
> **Answer for Typos Grammar Style And Presentation Improvements:** Thank you for raising the issue. We will make corrections in the next version.
>
> **Thank you again for your valuable feedback, and we hope that our responses address your concerns adequately.**

---

### Official Review · Reviewer_Gjy4 · 2023-08-04

**Soundness:** 3

**Excitement:**

3: Ambivalent: It has merits (e.g., it reports state-of-the-art results, the idea is nice), but there are key weaknesses (e.g., it describes incremental work), and it can significantly benefit from another round of revision. However, I won't object to accepting it if my co-reviewers champion it.

**Paper Topic And Main Contributions:**

This paper investigates representation disentanglement learning for end-to-end speech translation. A framework is proposed to disentangle speech into content representations and non-content representations based on cyclic reconstruction, feature reconstruction and speaker classification.
The main contribution is introducing representation disentanglement into end-to-end speech translation, to remove the non-linguistic information from the translation process.



**Questions For The Authors:**

It is helpful to add significantly analysis to Table 1 to show the significant improvements.
What does “MTL” stand for is not provided in the paper.

**Reasons To Accept:**

The proposed framework achieve superior performance than the compared baselines. The effectiveness of the masking, reconstruction, and speaker classification modules is verified with ablation studies.

**Reasons To Reject:**

The comparison with other disentanglement methods is absence. As the main contribution is the introduction of the disentanglement module to reduce non-linguistic information in the translation, demonstration of effectiveness of the disentanglement module is the most important, in addition to the evaluations at the very later text output performance.

**Reproducibility:**

4: Could mostly reproduce the results, but there may be some variation because of sample variance or minor variations in their interpretation of the protocol or method.

**Reviewer Confidence:**

4: Quite sure. I tried to check the important points carefully. It's unlikely, though conceivable, that I missed something that should affect my ratings.

---

> ### Author Rebuttal · Authors · 2023-08-29
>
> **We sincerely thank you for your detailed review and feedback. We will strive to integrate your constructive feedback into our paper.**
>
> **Reject_Reason**: Comparison with other disentanglement methods and demonstration of the effectiveness of the disentanglement module.
>
> **Answer for the Reject_Reason**: For comparison with other disentanglement methods, we INDEED have compared our approach with other disentanglement methods, the disentanglement method proposed by VQMIVC [4]. However, it yields worse performance and takes significantly more training time than CCSRD. It is important to acknowledge that the suitable disentanglement methods for E2E-ST are extremely limited. We hence don’t have many options to compare. This is because many disentanglement methods need supervised data for training, which is not feasible for E2E ST. Additionally, several of the remaining methods rely on specific complex modules or pre-processed features like pitch for training [1,2,3]. These methods do not align with our framework and, if adopted, would introduce substantially additional training and inference time.
>
> For the demonstration of the effectiveness of the disentanglement module, we have conducted analysis in Section 5. As the disentanglement module is designed to separate the content information from speech, we visually compared the speech content information and text embeddings in Figure 2. This visualization clearly illustrates that the disentangled speech representation closely resembles textual information, thus verifying the efficacy of the module.  Moreover, we conducted experiments under simulated noisy conditions, and the results shown in Table 3 provide further evidence of the module's effectiveness in disentangling speech. The cross-attention visualization experiments in Figure 3 also provide evidence that the decomposed content representation aligns more effectively with the target text content. Furthermore, the main experiment highlighted in Table 1 further reinforces the benefits of acquiring disentangled speech representations, which demonstrates better translation results can be achieved by leveraging disentangled content representations. These observations solidify the positive impact and significance of the disentanglement module on enhancing speech translation quality.
>
> **Question**: Significant analysis and the meaning of MTL.
>
> **Answer for the Question**: Many thanks for this suggestion. We have conducted significance tests on our experimental results and provide the details in Table 1. Notably, the ** symbol means the improvements over baseline is statistically significant (p < 0.01) and the results clearly demonstrate the significant improvements achieved by our method.
>
> Table 1: BLEU scores s of CCSRD under different settings on the MuST-C tst-COMMON set. ** means the improvements over W2V2-Transformer baseline is statistically significant (p < 0.01).
>
> |     Model     |     En-De     |     En-Es     |     En-Ru     |     En-Fr     |     En-It     |
> |   :-----   |    :-----:   |    :-----:   |    :-----:   |    :-----:   |    :-----:   |
> |     CCSRD-ST     |25.4**|30.2**|16.9**|35.8**|25.8**|
> |     CCSRD-MTL     |26.1**|31.0**|17.8**|37.1**|26.4**|
>
> Regarding MTL, it stands for multi-task learning, which is widely used in E2E-ST to further enhance translation performance. In our paper, the specific approach of MTL involves integrating additional MT and ASR tasks to the ST model using the same speech triplets, which is explained in line 230\~246 and line 422\~426.
>
> **Thank you again for your valuable feedback, and we hope that our responses address your concerns adequately.**
>
> [1]Towards Disentangled Speech Representations (Peyser et al., Interspeech 2022)\
> [2]Disentangled Speaker Embedding for Robust Speaker Verification (YI et al., ICASSP 2022)\
> [3]SpeechSplit2.0 Unsupervised Speech Disentanglement for Voice Conversion without Tuning Autoencoder Bottlenecks (Chan et al., ICASSP 2022)\
> [4]VQMIVC Vector Quantization and Mutual Information-Based Unsupervised (Wang et al., Interspeech 2021)

---

### Official Review · Reviewer_GTrW · 2023-08-06

**Soundness:** 2

**Excitement:**

3: Ambivalent: It has merits (e.g., it reports state-of-the-art results, the idea is nice), but there are key weaknesses (e.g., it describes incremental work), and it can significantly benefit from another round of revision. However, I won't object to accepting it if my co-reviewers champion it.

**Paper Topic And Main Contributions:**

This paper focus on end-to-end speech translation, which is the task of translating speech from one language to another without relying on intermediate transcriptions. The paper proposes a novel framework called CCSRD, which stands for Content-Centric Speech Representation Disentanglement. The main contributions of the paper are:

CCSRD learns to separate the linguistic content information from the non-linguistic speech factors (such as pitch, timbre, speaker identity, etc.) by using a disentanglement module that consists of cyclic reconstruction, feature reconstruction and speaker classifier tasks.
CCSRD does not require any transcription data for disentanglement learning, which makes it applicable to scenarios that lack or do not have transcription data at all.
CCSRD achieves an average improvement of +0.9 BLEU in two settings across five translation directions over the baseline, outperforming state-of-the-art end-to-end speech translation models and cascaded models.

**Reasons To Accept:**

This paper proposes a novel framework for end-to-end speech translation that learns to disentangle linguistic content from non-linguistic speech factors, such as pitch, timbre, and speaker identity. It does not require any transcription data for disentanglement learning, which makes it applicable to scenarios that lack or do not have transcription data at all. It achieves an average improvement of +0.9 BLEU in two settings across five translation directions over the baseline, outperforming state-of-the-art end-to-end speech translation models and cascaded models.

**Reasons To Reject:**

I suggest the authors to give the detailed analysis of the disentanglement module and how it affects the speech translation quality.

**Reproducibility:**

2: Would be hard pressed to reproduce the results. The contribution depends on data that are simply not available outside the author's institution or consortium; not enough details are provided.

**Reviewer Confidence:**

3: Pretty sure, but there's a chance I missed something. Although I have a good feel for this area in general, I did not carefully check the paper's details, e.g., the math, experimental design, or novelty.

---

> ### Author Rebuttal · Authors · 2023-08-29
>
> **We sincerely thank you for your detailed review and feedback. We will strive to integrate your constructive feedback into our paper.**
>
> **Reject_Reason:** Detailed analysis of the disentanglement module and how disentanglement module affects the speech translation quality.
>
> **Answer for the Reject_Reason:** First, we think we have done sufficiently detailed analysis on the disentanglement module. The analysis of the disentanglement module in our understanding should include architecture and function. For the architecture aspect, we refer to the main disentanglement module proposed in DiCyR (Bertoin and Rachelson, 2022), forcing the content and non-content encoders to learn content and non-content information through different tasks. The detailed network architecture is presented in Section 3.2 and 3.3. Furthermore, we have conducted an ablation study to analyze the effectiveness of different tasks within the module. For the function aspect, we have conducted the main analysis in Section 5. As the disentanglement module is designed to decompose representations, we performed the in-depth analysis on the obtained decomposed representations to verify the functionality of the module. In Figure 2, we have visualized the speech and text representations, demonstrating that the disentangled speech representation is closer to textual information. Moreover, we conducted experiments under simulated noisy conditions. The results presented in Table 4 show the successful disentanglement of speech by the disentanglement module.
>
> Second, the analysis in Section 5 also verifies the effectiveness of the disentanglement module in enhancing translation performance by obtaining high-quality textual content representation. This is supported by the visualization of speech and text representations in Figure 2, which clearly demonstrates the ability of the disentanglement module in separating content information. The denoising experiment presented in Table 4 further validates the module's ability to dissociate non-content content successfully. Furthermore, the cross-attention visualization experiments in Figure 3 provide evidence that the decomposed content representation aligns more effectively with the target text content. These experiments collectively support our motivation of utilizing the disentanglement module to facilitate the content encoder in learning superior content representations while mitigating the negative impact of non-content on translation.
>
> **Thank you again for your valuable feedback, and we hope that our response addresses your concerns adequately.**
>
> [1]Disentanglement by cyclic reconstruction.  (Bertoin and Rachelson, 2022)

---

### Meta-Review · Area_Chair_jpej · 2023-09-15

**Recommendation:** 3

**Metareview:**

Work is dedicated to end-to-end speech translation and proposes a framework (CCSRD) that disentangle linguistic content from non-linguistic content in the speech representations. The motivation for learning such disentangled representations is clear and method proposed is interesting. The method is evaluated on a speech2text translation task (MustC) and it achieves an improvement compared to baselines. However, reviewers highlighted the lack of detailed analysis of the disentanglement module and not enough comparisons with other approaches. Moreover, it is unfortunate that the work is focusing only on speech-to-text translation task and does not use the ‘non-content representation’ during inference as well (for instance for a S2S task or a speaker id task or emotion recognition task). As the non-content representation is unused, the paper sounds a bit incomplete and i’d recommend only an acceptance to findings.

---

### Decision · Program_Chairs · 2023-10-07

**Decision:**

Accept-Findings

**Comment:**

Work is dedicated to end-to-end speech translation and proposes a framework (CCSRD) that disentangle linguistic content from non-linguistic content in the speech representations. The motivation for learning such disentangled representations is clear and method proposed is interesting. The method is evaluated on a speech2text translation task (MustC) and it achieves an improvement compared to baselines. However, reviewers highlighted the lack of detailed analysis of the disentanglement module and not enough comparisons with other approaches. Moreover, it is unfortunate that the work is focusing only on speech-to-text translation task and does not use the ‘non-content representation’ during inference as well (for instance for a S2S task or a speaker id task or emotion recognition task). As the non-content representation is unused, the paper sounds a bit incomplete and i’d recommend only an acceptance to findings.